# Enacted Stigma in Inflammatory Bowel Disease: An Italian Phenomenological Study

**DOI:** 10.3390/healthcare11040474

**Published:** 2023-02-07

**Authors:** Silvio Simeone, Caterina Mercuri, Cristina Cosco, Vincenzo Bosco, Caterina Pagliuso, Patrizia Doldo

**Affiliations:** Department of Experimental and Clinical Medicine, Magna Graecia University of Catanzaro, 88100 Catanzaro, Italy

**Keywords:** stigma, phenomenological study, inflammatory bowel disease, enacted stigma

## Abstract

Inflammatory bowel disease (IBD) is a chronic disease of the gastrointestinal tract that has a profound impact on the quality of life of those afflicted with it. The scientific literature shows how the quality of life of people with IBD influences and is influenced by the clinical manifestations of the disease. Strongly connected with excretory functions, which have always been taboo in society, these clinical manifestations can lead to stigmatizing behaviours. The purpose of this study was to understand the lived experiences of the enacted stigma of people with IBD through Cohen’s phenomenological method. Two main themes (stigma in the workplace and stigma in social life) and one subtheme (stigma in love life) emerged from the data analysis. The data analysis revealed that stigma is associated with a multitude of negative health outcomes for the people targeted by it and can add to the already complex physical, psychological and social burdens endured by people with IBD. Having a better understanding of the stigma attached to IBD will facilitate the development of care and training interventions that can improve the quality of life of people suffering from IBD.

## 1. Introduction

Inflammatory bowel disease (IBD), such as Crohn’s disease (CD) or ulcerative colitis (UC), is a lifelong, chronic, progressive inflammatory disease of the gastrointestinal tract with an increasing incidence in industrialised countries [1]. It is characterised by an unpredictable course, with periodic fluctuations between flare-ups and remission [2]. Abdominal pain, bloody diarrhoea, weight loss and chronic fatigue are the symptoms most commonly described by IBD patients [3], often associated with extraintestinal manifestations involving the joints (peripheral and axial arthropathies), skin (erythemanodosum), eyes (episcleritis) and liver [4].

IBD profoundly affects patients’ health and quality of life (QoL) [3,5] and has negative psychological effects [3,6]. The interactions between the physical and psychological consequences of IBD have been documented [7,8,9]. IBD patients have very high rates of anxiety and depression compared to the healthy population [10]. Anxiety and depression are associated with more frequent disease flare-ups [11] and a more aggressive presentation [12]. Furthermore, stigma can add to the already complex physical, psychological and social burdens borne by people with IBD [13].

Stigma is a powerful social process in which one is labelled with negative attributes and stereotypes that lead to discrimination and loss of social status [14]. Health-related stigma can be defined as the social exclusion of a person or group of people with a particular health condition due to a medically unjustified adverse social judgment [13]. This type of stigma has mainly been investigated in relation to mental health and chronic disease conditions [7,15]. However, stigma has also been attached to IBD [7,16,17].

Associated with a multitude of negative health outcomes for the people targeted by it [15], stigma is multifaceted and can be conceptualised in terms of perceived, internalised and emanated stigma. Perceived stigma refers to individuals’ perception of others as having negative attitudes toward them. Internalised stigma refers to the acceptance of such attitudes as part of one’s identity. Emanated stigma refers to the direct experience of social discrimination due to one’s condition [18]. Each type of stigma has an impact on the perceived QoL of the targeted individual [7].

Recent literature reviews have shown that there have been few studies on stigmatisation from the 2000s to the present, suggesting the need to conduct further studies investigating particularly IBD-related stigma and showing how stigma, in its various forms, affects the perceived QoL of people with IBD, especially in relation to the social and psychological dimensions [7,19]. A better understanding of the stigma linked to other chronic diseases has allowed a better understanding of the lived experiences of those afflicted with such diseases, which has facilitated the development of training interventions that have improved their QoL [20,21].

Given the characteristics of IBD, its stigmatisation differs from that of other health conditions [22]. Furthermore, IBD-related emanated stigma has been less investigated worldwide than IBD-related perceived and internalised stigma [7,15,23]. The literature also recommends that studies exploring IBD-related stigma be conducted [17,19]. Expanding the understanding of IBD-related stigma can contribute to the development of strategies to reduce the stigma and improve the perceived QoL of patients with IBD. Therefore, the purpose of this study was to explore the lived experiences of the stigma of Italians with IBD, particularly their direct experiences of social discrimination on account of their health condition.

## 2. Materials and Methods

### 2.1. Design

The present study was conducted using Cohen’s phenomenological method [24]. This method, which combines descriptive (Husserlian) and interpretative (Gadamerian) phenomenology, has been successfully used in previous studies [25,26]. Cohen’s method was selected due to its ability to generate a deeper understanding of both lived experiences and the meanings attributed to them [27]. As Cohen et al. [24] stated, ‘The meanings that participants attribute to their experiences help create the needs they have and how these needs can best be met’. While the descriptive phenomenological method seeks to describe the conscious everyday experiences of the participants, putting any preconceived opinions in parentheses [28], the interpretive phenomenological method involves a detailed examination of the personal lived experiences of the participants.

This study followed the Consolidated Criteria for Reporting Qualitative Studies checklist as its standard [29].

### 2.2. Sample

The enrolment of study participants took place through convenience sampling. The Operative Unit of Gastroenterology and Interventional Endoscopy of Mater Domini Hospital, University of Magna Graecia, Catanzaro, helped us establish contact with individuals afflicted with IBD. The potential participants were first contacted via telephone to explain to them the nature and purpose of the study and to invite them to participate in it. Those who gave their consent to participate in the study were subsequently contacted again to agree on the date, time and place of the interview. Before proceeding with the interviews, the researcher, who had not had any previous contact with the participants, again explained to them the nature and purpose of the study. It was also made clear to them that they could withdraw from the study at any time and that the data to be obtained from them would be kept confidential at all stages of the study. All the participants then signed the informed consent form.

Out of a total of 17 subjects contacted, 3 refused to participate in the study for personal reasons; none of the subjects who participated in the study withdrew.

To be enrolled in the study, each prospective participant had to meet the following inclusion criteria: (1) had been diagnosed with IBD (i.e., Crohn’s disease or ulcerative colitis) based on clinical, endoscopic, histological and/or radiological evaluation; (2) of legal majority age under Italian law; (3) understood and spoke the Italian language correctly; (4) agreed to participate in the study by signing the informed consent form. Individuals with severe and pre-existing cognitive deficits (e.g., dementia) or cancer (terminal or actively being treated) and/or severe organ failure known to be associated with poor QoL were excluded from the study. The data pertaining to any subject who would later withdraw from the study were to be excluded from the analysis.

### 2.3. Data Collection

All the participants chose the date, time and place of their respective interviews. This helped them truthfully recount and describe their experiences. Before the data collection, all the researchers ‘bracketed’ or ‘put in brackets their own ideas and preconceptions under consideration’. This helped the researchers prevent their ideas from influencing the data analysis, thus improving the credibility of the study results [24].

The interviews were conducted with a single ‘open question’ to guarantee the participants’ maximum freedom of expression [30].

In each interview, the researcher/interviewer first defined ‘stigma’ and then asked the interviewee to describe their experiences involving the stigma directed at them. The researcher/interviewer maintained a ‘welcoming attitude’ 2 [31,32] throughout the interview, made ‘field notes’ and paid attention to the context of each statement by the interviewee and the interviewee’s ‘non-verbal language’.

When it seemed that the interviewee had nothing more to say, the researcher/interviewer ended the interview. Each interview was audio-recorded (with the permission of the interviewee) and was 20–60 min long.

The sampling and interviews continued until ‘data saturation’ was reached or when the themes that were extracted were already redundant [30]. Data saturation was reached after 14 interviews.

So that the researchers would become familiar with the chosen research method, two interviews that were not included in the study were conducted, and the data obtained from them were analysed.

### 2.4. Data Analysis

Each interview was faithfully transcribed verbatim and integrated with the researchers’ field notes. The researchers read all the transcripts initially to obtain an overview of the participants’ experiences expressed during the interviews. Subsequently, the researchers reread the transcripts line by line, assigning indicative themes to various passages. They then compared the extracted themes and did not find any discrepancies. The participants subsequently confirmed these themes, again without any discrepancies.

### 2.5. Rigour

Compliance with Lincoln and Guba’s (1985) criteria further confirmed the scientific rigour of the present study.

Continuous sampling up to data saturation ensured credibility. For the present study, credibility corresponded to internal validity and was the criterion for ensuring that the study examined what it was supposed to examine.

Reliability was guaranteed by the triangulation technique used in the study. The conduct of the data analysis by two or more researchers allowed us to confirm the results from different perspectives.

Confirmability was guaranteed by the briefing of the participants and their confirmation of the themes extracted from the data obtained from them.

Transferability was ensured through the participants’ in-depth descriptions of their experiences and the description of their sociodemographic characteristics.

### 2.6. Ethical Considerations

Ethical clearance for this study was obtained from the Ethics Committee of the Calabria Region Centre on 21 April 2022 (Re.Prot.n.119). The study was conducted according to the guidelines of the Declaration of Helsinki.

## 3. Results

Our sample consisted of 14 subjects (8 males, 6 females) with an average age of 39 years. Most had a high school education, and almost all were married or cohabited (Table 1). Two main themes emerged from the analysis of the interviews: stigma in the workplace and stigma in social life. The latter incorporated the subtheme stigma in love life.

### 3.1. THEME 1: Stigma in the Workplace

All the participants had well-described episodes of stigmatisation within their work context. That is, they described episodes of stigmatisation that emanated from or were attributable to their workplaces. The situation was characterised as or considered friendly up to the moment of stigmatisation, which was followed by a sense of isolation.

[Sitting in front of us, the serious face hinting at an almost incredulous smile, wide-open eyes and an open right hand rolling over the shoulders] I clearly saw that all my colleagues avoided going to the bathroom after me, or rather to my bathroom…One day, one of them, seeing that the second bathroom in the workplace was occupied, preferred to enter the men’s room, which was free at that time, rather than the one I had just used…Yet, we all do it [smiles a little more, settles back on the chair]…Even touching something touched by me is now avoided…One day, they put cans of deodorant on the counter. After I touched them, they just left them there…not them they took off even if it was their turn to fix…well, as they say…when I go, I only work there, just for me….(CX03)

I remember well the sense of friendship that was there at first, then slowly, due to my illness…Let’s say that COVID, or rather the protection regulations, was not new to me. My colleagues previously did not understand my behaviour because they did not know about my disease, but once they got to know about it, they started to look strangely at me from a distance…They washed their hands every time they touched something touched by me…They started to avoid having coffee with me.(EV05)

My colleagues avoid staying in the lift with me…When I enter and they’re already in the lift, they always pretend that they have to retrieve something or want to use the stairs instead for some reason…Now, I don’t care. I know I have to go there and work…They’re only my colleagues, not my friends. Not everyone can be your friend.(KP11)

### 3.2. THEME 2: Stigma in Social Life

It is clear how the participants’ social lives were changed by people’s stigmatising attitudes towards them. This main theme of the study data included the subtheme of stigma in love life; in fact, the participants clearly separated their networks of friends from those related to their intimate sphere and showed how stigmatising behaviours that influenced their overall experiences were also present in this aspect of their lives.

My friend called me and said, ‘We’ve organised an evening for us, just to be together for a while…Sorry, you can’t come because of the food. We’re telling you only now because we didn’t want to embarrass you…It would be embarrassing if you’d brought food to the pub from outside. I don’t think you can even think of doing such a thing’…Then, you find yourself not looking for them anymore.(DW04)

[With a serious, sad face, he turns his gaze to the window illuminated by sunlight.] They call you on the phone, yes, but if they’re to go out for dinner, they don’t call you, or worse, they tell you that they’ve already left and that they didn’t ask you if you wanted to come because they already knew what you would answer…Even at official meetings, such as at school, during interviews with the teachers…well, they don’t socialise with you, they tend to exclude you… Worse, when you go out of the council room, they offer a packet of handkerchiefs to you in front of everyone…assuming you’ll go to the bathroom, that you’ll….(BY02)

#### SUBTHEME 2.1: Stigma in Love Life

Even the participants’ private lives, specifically the sentimental sphere, had been the subject of clearly stigmatising behaviours and actions. The younger participants reported clear stigmatising episodes involving their partners and how these behaviours affected them.

Well, let’s face it. It’s not easy to accept a girl…especially at the beginning…in short, the image of being intimate with her and she suddenly has to run to the bathroom…or worse…it’s not the ideal image…[She blushes slightly, lets out a nervous smile and rubs her hands together. One fist rubs the inside of the other hand. Sitting on the chair, she moves slightly, bringing her torso forward. She lowers her voice, brings her knees together, with her legs forming a triangle-like angle]…Fortunately, she then meets the right person and, over time, she gets to trust him and tells him…well, even if it can escape everyone [smiles blushing even more] or at least it can happen…with the right person then, after the beginning, we also clarify many aspects that seem strange…I always speak referring to intimacy…an aspect perhaps neglected by those who give care for us but important for our lives.(HS08)

It’s not easy at all. You see, if it happens, in short, you suddenly have to run to the bathroom and odours and noises come out, well [stiff on the chair, broad shoulders and back very straight, forming a 90° angle with the seat; the palm of the right hand nervously rubs the jeans on the right thigh, while the look on the face is serious. The legs, crossed under the chair, move nervously]… She erases you because she doesn’t know…For her, you no longer exist…then you see that the others look at you strange, from a distance…that then everyone they do, noh?! [a smile appears on the face, the hand, for a moment, stops rubbing on the thigh]…maybe it should be a business card, show up immediately and see if she is able to resist the image that one is done….(FU06)

## 4. Discussion

Our study investigated lived experiences related to the enacted stigma towards Italian subjects suffering from IBD. To our knowledge, this was the first study to use a phenomenological method to explore how enacted stigma affects the lives of people with IBD.

From the analysis of the scientific literature, it appears that people with IBD tend to be stigmatised to the extent that they experience symptoms that are commonly seen as disgusting [23]. Enacted stigma is the least investigated, probably because the behaviours suffered are not immediately perceived as important [23,33]. This type of stigma is more frequent in the world of work, as shown by the participants’ narrations.

However, the aforementioned behaviours, whose memories are still quite vivid in the minds of the participants, contribute to the low levels of disclosure of the sufferers’ health conditions, thus hindering social support for them [34] and negatively affecting their self-perceived QoL [23].

The results of our data analysis were surprising as the scientific literature shows that the duration of a disease seems to diminish the feeling of having suffered stigmatising behaviours [35]. The participants in this study have well imprinted in their minds the stigmatizing behaviours suffered and perfectly remember how they felt about it. The lived experience of our participants shows, contrary to what is reported in the literature, that the duration of an illness does not seem to decrease the feeling of having undergone stigmatizing behaviours. Although they did not state precisely when they experienced the stigmatising behaviours, the effects of such behaviours on them were clear in their stories. Similar to teasing, enacted stigma has been minimally investigated and seems common, particularly among the young [7]. However, the narrations of the participants in the present study clearly show that some of these behaviours are they have been told by people who are no longer very young. In fact, they are all clearly attributable to the world of work, and comparing the age of the participants, we are not dealing with a very young population.

The second theme that emerged was linked to the stigma emanating from one’s social life. Stigma can lead to social isolation [9]. Almost all the participants reported how their friendships with people who manifested stigmatising behaviour ended [36]. Social relationships play a fundamental role in people’s lives [21], so it is clear how such behaviours can harm people with IBD. The two-way link between IBD and mental health, which makes the patients’ overall health more precarious, has become increasingly well known, with mental health recognised as affecting the presentation of IBD symptoms [37].

Culturally, everything associated with stool and bowel movements is considered ‘dirty’ and taboo [38]. Thus, it is important for health professionals to utilise additional resources to improve the QoL of people with IBD. Italian culture is highly linked to socialisation and food and could have influenced the participants’ stigma-related experiences. Thus, IBD-related stigma has far-reaching implications for health professionals caring for people with IBD or other chronic illnesses and for their families [39]. The correct identification of patients at risk can allow adequate support to be offered, accelerating the path to disease acceptance and stigma reduction [21]. For this reason, further specific studies are recommended [7], especially regarding the areas highlighted by the participants in the present study as those where support is needed.

The theme that emerged linked to one’s relational and sexual life is truly a novelty in the literature. In his book Stigma: Notes on the Management of Spoiled Identity, Goffman [40] explains that the stigmatised person is not really a person but a social situation generated by behavioural norms. This highlights the importance of health professionals’ educational role. This educational role was explicitly requested by the participants in the present study as they believe that it could conserve their private or intimate life. They recognised that the absence of educational interventions threatens their QoL, as reported by other researchers [41]. Although no participants described changes in their sex life, the request to address this topic was direct. The intimacy of a couple could be complicated because, in [41] addition to fatigue and intestinal symptoms, even physical contact arouses fear, while there seems to be a need for ‘cuddling’ [42]. This need for intimacy and affection can lead to an alteration of the couple’s relationship, altering the perception of one’s quality of life. As reported in a very recent qualitative study investigating the sexual health of subjects with IBD [41], regardless of whether one finds themselves living with IBD in a phase of life in which sexual identity emerges, or at the peak of period of conception or in older age, the need for this educational approach is strongly felt. In fact, despite an Italian culture typically closed to certain topics, the request was explicit.

Another novelty of our results is the surprising explicit request, also received from women, which contrasts with what is shown by the international literature. Indeed, studies show how difficult it is for women to address issues related to their sex life with people outside an intimate family context (e.g., healthcare professionals) [42]. The cultural heritage of our sample probably had a strong influence on those who disclosed information related to their sexual health. The participants made numerous references to and jokes about the intellectual functionality common to all living beings. The constant comparison of IBD symptoms with natural intestinal functions is a theme present in the literature; we tend to use humorous expressions, reiterating that bowel functions are the same for everyone as a coping strategy [22].

Even the incidents of reluctance to touch objects or materials owned or touched by the participants can be said to reflect a lack of knowledge and to mirror the stigmatising experiences linked to infectious diseases [43]. This demonstrates how poor general knowledge of a topic can inevitably lead to stigmatising and harmful behaviour. The results of an experimental study showed a positive correlation between an increase in knowledge of IBD and a reduction in the stigma attached to it [44].

The present study had some limitations. First, the sample consisted of people with IBD coming from a single territorial area. Thus, the social and cultural differences within the national territory may not have been represented by the study sample. The sample size, obtained when the data were saturated, cannot be considered a limit for the type of study. However, despite these study limitations, the fact that the study focused on IBD-related stigma shows a willingness to explore it, starting with the people afflicted with IBD in Italy. As such, we believe that our study jumpstarts much-needed efforts to understand the experiential reality of people with IBD.

## 5. Conclusions

This study provided important information on the enacted stigma experienced by people with IBD. Enacted stigma may have considerable repercussions on the QoL of these people and may also worsen the clinical signs and symptoms of the disease. To our knowledge, this study was the first phenomenological study on IBD-related enacted stigma conducted in Italian territory; therefore, its results can help guarantee true holistic assistance to people with IBD by fostering an understanding of the lived experiences of such people and by laying the foundations for effective educational interventions.

## Figures and Tables

**Table 1 healthcare-11-00474-t001:** Sample’s characteristics.

Code.	Gender	Age	Marital Status	No. of Kids	Education	Employment	Diagnose	Diagnostic Time	Disease Phase	Treatment
AZ 01	Male	32	Single	-	High School	Casual Worker	CD	8 years ago	Flare-up	Infliximab
BY 02	Female	35	Married	2	Middle School	Other	UC	14 years ago	Remission	Infliximab
CX 03	Female	22	Married	-	High School	Worker	CD	5 years ago	Remission	Vedolizumab
DW 04	Male	28	cohabitant	1	High School	Self-Worker	CD	7 years ago	Remission	Vedolizumab
EV05	Male	47	Married	3	High School	Worker	CD	25 years ago	Flare-up	Infliximab
FU 06	Male	26	cohabitant	-	High School	Self-Worker	CD	7 years ago	Remission	Vedolizumab
GT 07	Male	45	Single	--	High School	Worker	UC	12 years ago	Remission	Infliximab
HS 08	Female	32	Married	2	High School	Self-Worker	UC	9 years ago	Remission	Mesalazina-5-ASA
IR 09	Female	35	Married	3	Middle School	Worker	CD	9 years ago	Flare-up	Mesalazina
JQ 10	Female	43	Single	-	Graduation	Self-Worker	UC	28 years ago	Remission	Infliximab
KP 11	Male	47	Married	2	Master’sDegree	Worker	CD	20 years ago	Flare-up	Infliximab
LO 12	Female	51	Married	2	Graduation	Self-Worker	UC	4 years ago	Flare-up	Infliximab
ML 13	Male	53	Married	2	High School	Worker	UC	28 years ago	Remission	Mesalazina-5-ASA
NM 14	Male	52	Married	2	High School	Worker	CD	24 years ago	Remission	Vedolizumab

## Data Availability

The interviews transcribed and analysed for this study are not publicly available. However, this could be shared by the authors upon reasonable request and with permission of the University Magna Graecia of Catanzaro.

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
