# Peer review of "Enacted Stigma in Inflammatory Bowel Disease: An Italian Phenomenological Study"

_healthcare, 2023, doi:10.3390/healthcare11040474_

Round 1

Reviewer 1 Report

Comments and suggestions for Authors

The manuscript entitled ‘Enacted stigma in inflammatory bowel disease: an Italian phenomenological study’ presents the lived experiences in terms of stigmatization due to their medical condition.

Introduction:  is clearly and concisely written.

Method and materials:

Regarding the design, the authors wrote that the study used the Cohen’s phenomenological method which ‘has been successfully used in previous studies [25-27]’ (line 72). But the mentioned studies were carried out by the same author as that of the present study. The authors should also specify other studies and other authors to support this statement.

The authors should also mention other aspects related to the sample: which was the response rate (how many potential patients were contacted and how many agreed to participate), how many have withdrawn from the study (if any).

Lines 124-126 – the paragraph refers to an aspect that took place before the start of the actual study, perhaps it should be written at the beginning of the Data collection section.

Results:

I appreciate that the authors paid attention to the interviewee’s ‘non-verbal language’ and they included these observations into the manuscript, along with the fragments of participants’ descriptions.

Discussions:

The authors stated that ‘To our knowledge, this was the first study to use a phenomenological method to explore how enacted stigma affects the lives of people with IBD’ (lines 230-231). But, according to the manuscript references list, there were other studies that investigated this aspect. Perhaps, it was about Italian patients, but in this case the manuscript would have gained if it revealed similarities with other studies or some particular aspects.

References

Out of 43 references, approximately half (24) are from the last 5-6 years.

The authors should reconsider to what extent self-citation of 3 articles is appropriate

Although I appreciate the work of the authors, the article does not bring anything significant new and does not substantially contribute to the prior literature on this topic. As such, I cannot recommend publication of this article in Healthcare.

28.11.2022

Reviewer 2 Report

I ask the authors to make the following changes/corrections:

1. In R153, the text in brackets should be excluded. In a series of 14 cases, it is not correct to calculate the SD.

2. The text needs to be corrected because there are some typing mistakes.

3. The bibliography must be written according to MDPI recommendations.

I congratulate the authors for their work!

The study should be continued. If a group of at least 28-30 subjects can be formed, you should also do a statistical processing of the data. I suggest you use the corrected Yates formulas for small groups.

Reviewer 3 Report

Thanks for inviting me to review this interesting paper on the social stigma related to inflammatory bowel diseases. Although this paper identified a critical area of research and provided a phenomenological perspective, I found the soundness of the qualitative research insufficient to support it.  Major concerns are listed below. 

1. The representativeness of the sample should be justified. 

2. How the disease severity affects the conclusion should be articulated in the paper. 

3. I wonder whether the interview is structured or semi-structured. I may have slight concerns that these participants were guided toward the lived stigma that may not necessarily be exclusive to IBD. The authors should clarify the methods. in more detail.

Round 2

Reviewer 1 Report

Authors have addressed some of the comments and suggestions which I have made in my previous review.

Still, it is not sufficiently presented what is the novelty of this study, what this study adds to the prior literature.

In the Discussions section (which remained the same as in the first version) the authors should add much more clearly the particularities of their study, what they identified to be specific to the studied population. There are only two ideas (related to the duration of the disease and to the participants’ age), but the authors should elaborate more to convince about the scientific significance of this study.

25.01.2023

Reviewer 3 Report

My concerns have been largely addressed. I have no more questions. 
